# Do Language Models Understand Time?

Xi Ding
Australian National University
Canberra, Australian Capital Territory, Australia
Xi.Ding1@anu.edu.au

Lei Wang*
Griffith University
Brisbane, Queensland, Australia
Australian National University
Canberra, Australian Capital Territory, Australia
l.wang4@griffith.edu.au

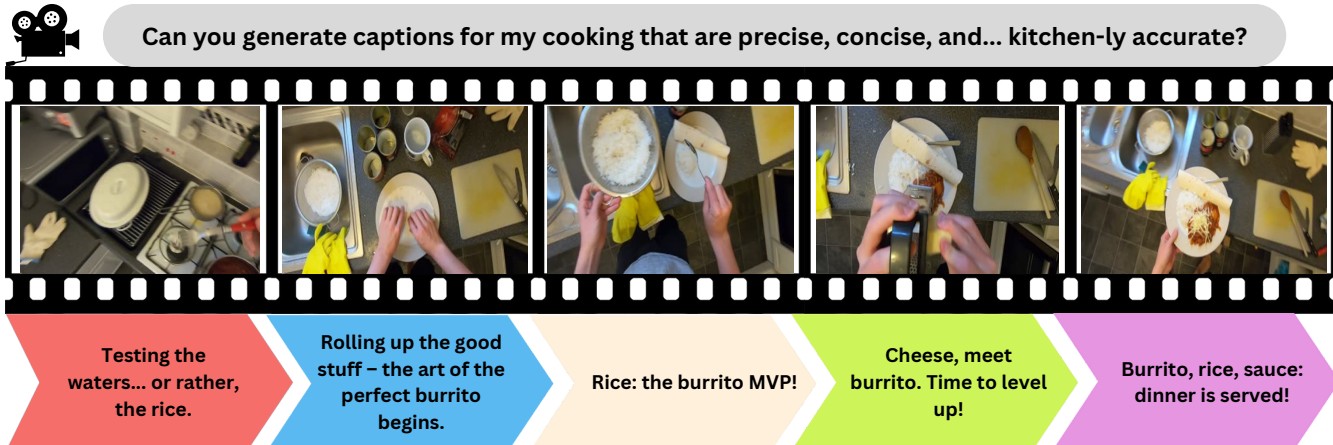

Can you generate captions for my cooking that are precise, concise, and... kitchen-ly accurate?

Testing the waters... or rather, the rice.

Rolling up the good stuff – the art of the perfect burrito begins.

Rice: the burrito MVP!

Cheese, meet burrito. Time to level up!

Burrito, rice, sauce: dinner is served!

Figure 1: Do language models understand time? In the kitchen arena, where burritos are rolled, rice waits patiently, and sauce steals the spotlight, LLMs try their best to keep up. Captions flow like a recipe—precise and tempting—but can they truly tell the difference between prepping, cooking, and eating? After all, in cooking, timing isn't just everything—it's the secret sauce!

## Abstract

Large language models (LLMs) have revolutionized video-based computer vision applications, including action recognition, anomaly detection, and video summarization. Videos inherently pose unique challenges, combining spatial complexity with temporal dynamics that are absent in static images or textual data. Current approaches to video understanding with LLMs often rely on pretrained video encoders to extract spatiotemporal features and text encoders to capture semantic meaning. These representations are integrated within LLM frameworks, enabling multimodal reasoning across diverse video tasks. However, the critical question persists: Can LLMs truly understand the concept of time, and how effectively can they reason about temporal relationships in videos? This work critically examines the role of LLMs in video processing, with a specific focus on their temporal reasoning capabilities. We identify key limitations in the interaction between LLMs and pretrained encoders, revealing gaps in their ability to model long-term dependencies and abstract temporal concepts such as causality and event progression. Furthermore, we analyze challenges posed by existing video datasets, including biases, lack of temporal annotations, and domain-specific limitations that constrain the temporal understanding of LLMs. To address these gaps, we explore promising future directions, including the co-evolution of LLMs and encoders, the development of enriched datasets with explicit temporal labels, and innovative architectures for integrating spatial, temporal, and semantic reasoning. By addressing these challenges, we aim to advance the temporal comprehension of LLMs, unlocking their full potential in video analysis and beyond. Our paper's GitHub repository can be found here.

## CCS Concepts

• **Computing methodologies** → **Motion capture**; *Neural networks*; • **Information systems** → *Language models*; World Wide Web.

## Keywords

Language language models, Videos, Temporal, Interaction

**ACM Reference Format:**
Xi Ding and Lei Wang. 2025. Do Language Models Understand Time?. In *Companion Proceedings of the ACM Web Conference 2025 (WWW Companion '25), April 28-May 2, 2025, Sydney, NSW, Australia.* ACM, New York, NY, USA, 14 pages. https://doi.org/10.1145/3701716.3717744

---

*Corresponding author.

# 1 Introduction

Large language models (LLMs) have brought transformative advancements to artificial intelligence (AI), excelling across a wide array of tasks in natural language processing and computer vision [11, 53, 186]. Their ability to understand and generate human-like language has enabled groundbreaking applications, from machine translation to image and video captioning [83] (see Figure 1, frames from EPIC-KITCHENS-100 [39]). More recently, the integration of LLMs into video processing has sparked significant interest, leading to advances in tasks such as action recognition [177, 178], anomaly detection [160, 202, 212], and video summarization [74, 104, 193, 213]. However, videos pose unique challenges compared to other modalities due to their dual reliance on both spatial and temporal information [21]. Unlike static images, videos capture the dimension of time, embedding sequential dynamics that demand sophisticated reasoning [101, 149]. Similarly, unlike textual data, videos involve rich, complex visual elements that require intricate modeling [26, 170].

Despite these advancements, a fundamental question remains unresolved: Do language models truly understand the concept of time? Temporal reasoning, the ability to comprehend and infer relationships between events over time, is essential for video-based tasks such as causal inference [23, 100], event prediction [63, 72], and understanding action progression [157]. While pretrained video encoders provide LLMs with spatiotemporal embeddings and text encoders contribute semantic insights, the fusion of these components often lacks the nuanced understanding of time required for more advanced applications [44]. Current methods rely heavily on pretrained encoders and dataset-specific tuning [21, 26, 121, 170], raising questions about the generalizability and scalability of these approaches. Table 1 provides a summary of recent video-LLMs, detailing the visual encoders they use and their mechanisms for interacting with these encoders.

This work aims to address these gaps and critically examines the role of LLMs in understanding temporal dynamics in video data. We focus on the interplay between visual (image/video) encoders and LLMs, exploring how effectively they bridge the gap between raw spatiotemporal features and high-level temporal reasoning. By tackling these issues, we seek to shed light on the limitations of existing approaches and inspire innovations in LLM-based video understanding. Our **contributions** are threefold:

i. We provide a detailed review of LLM applications in video processing, with a particular focus on their ability to comprehend temporal concepts, highlighting the state of the art and identifying key limitations.

ii. We analyze the shortcomings of existing LLM-based approaches, particularly their reliance on pretrained encoders and the challenges posed by video datasets, such as lack of temporal annotations and biases towards short-term dependencies.

iii. We propose actionable pathways for advancing LLMs' temporal understanding, emphasizing joint training strategies, better dataset design, and improved alignment between spatiotemporal and semantic features.

By addressing these aspects, this paper seeks to advance our understanding of temporal reasoning in LLMs and pave the way for more robust, generalizable, and scalable solutions in video analysis.

The insights offered here aim to engage researchers and practitioners alike, highlighting the importance of bridging the gap between static representations and dynamic reasoning in AI systems.

# 2 Related Work

The application of LLMs to video processing has attracted significant attention, owing to their capacity to bridge visual and textual modalities [173]. In this section, we review related work across four key areas: LLMs for video understanding, pretrained visual encoders, datasets for video understanding, and temporal reasoning in AI systems. We also highlight the distinct contributions of our work compared to prior studies.

**LLMs for video understanding.** LLMs have shown remarkable versatility in video-related tasks by incorporating multimodal learning frameworks [184]. Notable works, such as Flamingo [5] by DeepMind, integrate visual and textual modalities for tasks like video captioning and video question answering (QA). Flamingo uses a cross-modal attention mechanism to align spatiotemporal video embeddings with text representations, showcasing the potential of LLMs in multimodal fusion. Other models [55, 57, 153, 197], including OmniVL [153] and Florence [197], explore unified architectures that handle images, videos, and text simultaneously, reducing reliance on domain-specific encoders [84, 161–169]. However, these works primarily focus on improving task performance without a deep analysis of how LLMs handle temporal dynamics, leaving their capacity for explicit time reasoning largely unexamined.

While prior studies [7, 43, 64, 76, 141, 154] primarily emphasize task performance, we specifically investigate whether LLMs truly understand the concept of time. Our work explores the interplay between spatiotemporal embeddings and LLM frameworks, providing a deeper analysis of their temporal reasoning capabilities.

**Pretrained visual encoders in multimodal learning.** The success of LLMs in video understanding often hinges on the use of pretrained visual encoders [31, 51, 99, 174]. Models like CLIP [122], ResNet [68], and Vision Transformers (ViT) [45] are frequently used for spatial feature extraction, while video-specific encoders such as I3D [18], SlowFast [48], TimeSformer [9], and Video Swin Transformer [109] extract spatiotemporal features. These encoders are trained on large-scale datasets like ImageNet [41, 127] and Kinetics [16, 17, 82], enabling them to capture fine-grained features for downstream tasks. Table 2 presents an overview of widely used pretrained visual encoders along with their corresponding training datasets. Figure 2 compares the performance of visual encoders, showcasing image encoders evaluated on ImageNet-1K[41] and video encoders assessed on Kinetics-400 and Something-Something V2[58]. While the modularity of these encoders facilitates efficient system design, their reliance on general-purpose pretrained features poses limitations, particularly in domain-specific tasks and long-term temporal reasoning [84, 121, 161, 165, 166, 169]. Existing works often treat encoders as static components, overlooking the potential benefits of jointly optimizing encoders and LLMs for temporal understanding [133, 151].

Unlike works that use pretrained encoders as black-box components [134], we examine their limitations, including their bias

| Model | Venue | Visual encoder | Other modality encoders | Interaction / Fusion mechanism | Description |
|---|---|---|---|---|---|
| Flamingo [5] | NeurIPS 2022 | Normalizer-Free ResNet[10] | Text: Chinchilla[70] | Perceiver Resampler & Gated XATTN-DENSE | Visual-language model. |
| LaViLa [207] | CVPR 2022 | TimeSformer[9] | Text: 12-layer Transformer | Cross-attention modules | Large-scale language model. |
| mPLUG-2 [188] | ICML 2023 | CLIP ViT-L/14[122] | Text: BERT[42] | Universal layers & cross-attention modules | Modularized multi-modal foundation model. |
| Vid2Seq [191] | CVPR 2023 | CLIP ViT-L/14[122] | Text: T5-Base[124] | Cross-modal attention | Sequence-to-sequence video-language model. |
| Video-LLaMA [201] | EMNLP 2023 | CLIP ViT-G[122] | Text: Vicuna[35], Audio: ImageBind[54] | Aligned via Q-Formers for video and audio | Instruction-tuned multimodal model. |
| ChatVideo [151] | arXiv 2023 | e.g., OmniVL[153], InternVideo[178] | Text: ChatGPT[182], Audio: e.g., Whisper[122] | Tracklet-centric with ChatGPT reasoning | Chat-based video understanding system. |
| VideoChat [93] | arXiv 2023 | EVA-CLIP ViT-G/14 [140] | Text: StableVicuna[37], Audio: Whisper[122] | Q-Former bridges visual features to LLMs for reasoning | Chat-centric model. |
| VideoLLM [30] | arXiv 2023 | e.g., I3D[18], SlowFast [48] | Text: e.g., BERT[42], T5[124] | Semantic translator aligns visual and text encodings | Video sequence modeling using LLMs. |
| VAST [27] | NeurIPS 2023 | EVA-CLIP ViT-G/14[140] | Text: BERT[42], Audio: BEATs[28] | Cross-attention layers | Omni-modality foundational model. |
| Video-ChatGPT [115] | ACL 2023 | CLIP ViT-L/14 [122] | Text: Vicuna-v1.1[103] | Spatiotemporal features projected via linear layer | Integration of vision and language for video understanding. |
| Valley [112] | arXiv 2023 | CLIP ViT-L/14 [122] | Text: StableVicuna[37] | Projection layer | LLM for video understanding. |
| Macaw-LLM [113] | arXiv 2023 | CLIP ViT-B/16 [122] | Text: LLAMA-7B[146], Audio: Whisper[122] | Alignment module unifies multi-modal representations | Multimodal integration using image, audio, and video inputs. |
| Auto-AD II [65] | CVPR 2023 | CLIP ViT-B/32 [122] | Text: BERT[42] | Cross-attention layers | Movie description using vision and language. |
| Video-LLaVA [98] | arXiv 2023 | LanguageBind-Video [211] | Text: Vicuna v1.5[35] | MLP projection layer | Unified visual representation learning for video. |
| GPT4Video [180] | ACMMM 2023 | CLIP ViT-L/14 [122] | Text: LLaMA 2[146] | Transformer-based cross-attention layer | Video understanding with LLM-based reasoning. |
| LLaMA-VID [96] | ECCV 2023 | CLIP ViT-L/14 [122] | Text: Vicuna[35] | Context attention and linear projection | LLaMA-VID for visual-textual alignment in videos. |
| InternVideo2 [177] | ECCV 2023 | InternVL-6B[33], VideoMAE V2 [158] | Text: BERT-Large[42], Audio: BEATs[28] | Q-Former aligns multi-modal embeddings | Foundation model for multimodal video understanding. |
| COSMO [150] | arXiv 2024 | CLIP ViT-L/14 [122] | Text: OPT-IML[75]/RedPajama[145]Mistral[80] | Gated cross-attention | Contrastive-streamlined multimodal model. |
| VTimeLLM [72] | CVPR 2024 | CLIP ViT-L/14 [122] | Text: Vicuna[35] | Linear layer | Temporal video understanding enhanced with LLMs. |
| VILA [99] | CVPR 2024 | CLIP ViT-L/336px[122] | Text: LLaMA-2-7B/13B[146] | Linear layer | Vision-language model. |
| Video ReCap [74] | CVPR 2024 | TimeSformer [9] | Text: GPT-2[123] | Cross-attention layers | Recursive hierarchical captioning model. |
| OmniViD [152] | CVPR 2024 | VideoSwin [109] | Text: BART[91] | MQ-Former | Generative model for universal video understanding. |
| VTG-LLM [62] | arXiv 2024 | EVA-CLIP ViT-G/14[140] | Text: LLaMA-2-7B[146] | Projection layer | Enhanced video temporal grounding. |
| AutoAD III [66] | CVPR 2024 | EVA-CLIP ViT[140] | Text: GPT-3.5-turbo | Shared Q-Former | Video description enhancement with LLMs. |
| LAVAD [198] | CVPR 2024 | BLIP-2 ViT-L/14, ImageBind [54] | Text: Llama-2-13b-chat[146] | Converts video features into textual prompts for LLMs | Training-free video anomaly detection using LLMs. |
| MA-LMM [67] | CVPR 2024 | EVA-CLIP ViT-G/14 | Text: Vicuna[35] | A trainable Q-Former | Memory-augmented large multimodal model. |
| MiniGPT4-Video [6] | arXiv 2024 | EVA-CLIP ViT-G/14 [140] | Text: LLaMA 2[146] | Concatenates visual tokens and projects into LLM space | Video understanding with visual-textual token interleaving. |
| PLLaVA [190] | arXiv 2024 | CLIP ViT-L/14 [122] | Text: LLAMA-7B[146] | MM projector with adaptive pooling | Parameter-free extension for video captioning tasks. |
| V2Xum-LLaMA [71] | arXiv 2024 | CLIP ViT-L/14 [122] | Text: LLaMA 2[146] | Vision adapter | Video summarization using temporal prompt tuning. |
| VideoChat2 [94] | CVPR 2024 | UMT-L[107] | Text: Vicuna[35] | Linear projection | A comprehensive multi-modal video understanding benchmark. |
| MotionLLM [25] | arXiv 2024 | LanguageBind[211], VQ-VAE[204] | Text: Vicuna[35] | Modality translator: Motion / Video translator | Understanding human behaviors from human motions and videos. |
| VideoGPT+ [114] | arXiv 2024 | CLIP ViT-L/14[122], InternVideo2[177] | Text: Phi-3-Mini-3.8B[1] | MLP | Long-context video understanding. |
| EmoLLM [194] | arXiv 2024 | CLIP ViT-L/14[122] | Text: Vicuna-v1.5[35], Audio: Whisper[122] | Multi-perspective visual projection | Multimodal emotional understanding with improved reasoning. |
| Holmes-VAD [202] | arXiv 2024 | LanguageBind ViT-L/14[211] | Text: LLaMA3-Instruct-70B[4] | Temporal sampler | Multimodal LLM for video anomaly detection. |
| ShareGPT4Video [24] | arXiv 2024 | CLIP ViT-L/14[122] | Text: Mistral-7B-Instruct-v0.2[80] | MLP | Precise and detailed video captions with hierarchical prompts. |
| Vriptor [193] | arXiv 2024 | EVA CLIP ViT-L/14[140] | Text: ST-LLM[105], Audio: Whisper[122] | Scene-level sequential alignment | Vriptor for dense video captioning. |
| VideoLLaMA 2 [34] | arXiv 2024 | CLIP ViT-L/14[122] | Text: LLAMA 1.5[102], Audio: BEATs[28] | Spatial-Temporal Convolution (STC) connector | Advancing spatial-temporal modeling and audio understanding. |
| VideoLLM-online [22] | CVPR 2024 | CLIP ViT-L/14[122] | Text: Llama-2-Chat[146]/Llama-3-Instruct[4] | MLP projector | Online video large language model for streaming video. |
| Video-CCAM [47] | arXiv 2024 | SigLIP-SO400M[199] | Text: Phi-3-4k-instruct[1]/ Yi-1.5-9B-Chat[3] | Cross-attention-based projector | Causal cross-attention masks for short and long videos. |
| LongVA [205] | arXiv 2024 | CLIP ViT-L/336px [122] | Text: Qwen2-Extended[8, 192] | MLP | Long context video understanding. |
| InternLM-XComposer-2.5[203] | arXiv 2024 | CLIP ViT-L/14 [122] | Text: InternLM2-7B[15], Audio: Whisper[122] | MLP | Long-context LVLM supporting ultra-high-resolution video tasks. |
| VITA [50] | arXiv 2024 | InternViT-300M [31–33, 51] | Text: Mistral-8x7B [81], Audio: Mel Filter Bank | MLP | Open-source interactive multimodal LLM. |
| Kangaroo [104] | arXiv 2024 | EVA-CLIP-L [140] | Text: Llama-3-8B-Instruct[4] | Multi-modal projector | Video-language model supporting long-context video input. |
| Qwen2-VL [171] | arXiv 2024 | CLIP ViT-L/14[122] | Text: Qwen2-7B[8, 192] | Cross-attention modules | Vision-language model for multimodal tasks. |
| Oryx [108] | arXiv 2024 | OryxViT[108, 199] | Text: Qwen-7B/32B[8, 192] | Cross-attention | Spatial-temporal model for high-resolution understanding. |
| Video-XL [135] | arXiv 2024 | CLIP ViT-L[122] | Text: Qwen2-7B[8, 192] | Visual-language projector | Long-context video understanding model. |
| SlowFocus [119] | NeurIPS 2024 | CLIP ViT-L/14[122] | Text: Vicuna-7B v1.5[208] | Visual adapter (projector layer) | Fine-grained temporal understanding in video LLM. |
| VideoStudio [110] | ECCV 2024 | CLIP ViT-H/14 [122] | Text: CLIP ViT-H/14[122] | Cross-attention modules | Multi-scene video generation. |
| VideoINSTA [97] | arXiv 2024 | CLIP ViT-L [122] | Text: Llama-3-8B-Instruct[4] | Self-reflective spatial-temporal fusion | Zero-shot long video understanding model. |
| Loong [179] | arXiv 2024 | Causal 3D CNN[195] | Text: Standard text tokenizer | Decoder-only autoregressive LLM with causal attention | Autoregressive language models. |
| TRACE [63] | arXiv 2024 | CLIP ViT-L[122] | Text: Mistral-7B[80] | Task-interleaved sequence modeling & Adaptive head-switching | Video temporal grounding via causal event modeling. |
| Apollo[213] | arXiv 2024 | SigLIP-SO400M[199], InternVideo2[177] | Text: Qwen2.5-7B[8, 192] | Perceiver Resampler & Token Integration with Timestamps | Video understanding model. |

**Table 1: Summary of latest multimodal video-LLMs and their interaction / fusion mechanisms.**

| Type | Visual encoder | Pretrained dataset |
|---|---|---|
| Image | Normalizer-Free ResNet[10] | ImageNet-1K[41] |
| | CLIP ViT[122] | WebImageText[138] |
| | EVA-CLIP ViT[140] | LAION-2B[130], COYO-700M[12] |
| | BLIP-2 ViT[92] | WebImageText[138] |
| | SigLIP (e.g., ViT)[199] | WebLI[30] |
| | OryxViT[108, 199] | WebLI[30] |
| Video | TimeSformer[9] | Kinetics-400[82], Kinetics-600[16] |
| | I3D[18] | HMDB51[87], UCF101[137], Kinetics-400[82] |
| | SlowFast[48] | Kinetics-400[82], Kinetics-600[16], Charades[136] |
| | VideoSwin[109] | ImageNet-21K[127] |
| | UMT[107] | Kinetics-400[82], AudioSet[52] |
| | LanguageBind[211] | VIDAL-10M[211] |
| | VideoMAE V2[158] | Kinetics-400[82], -600[16], -700[17], Something-Something V2[58], AVA[60] |
| | InternVL (e.g., InternViT-6B) [32] | Hybrid image-text datasets (e.g., LAION-en)[14, 19, 61, 130, 131] |
| | InternVideo2[177] | Hybrid video datasets (e.g., Kinetics-400[82], InternVid[176]) |

**Table 2: Visual encoders with their pretrained datasets.**

toward short-term dependencies and their challenges in generalizing to abstract temporal concepts. We propose pathways for jointly optimizing encoders and LLMs to address these issues.

**Datasets for video understanding.** Datasets are fundamental to advancing video understanding, particularly for assessing the temporal reasoning abilities of LLMs [56, 77, 209]. Current datasets often focus on action recognition, video captioning, or question-answering, capturing spatiotemporal patterns and semantic connections [58, 78, 128]. However, many datasets emphasize short-term motions or provide only surface-level annotations, lacking temporal details such as event order, causality, or duration [82, 88, 137, 175].

While multimodal datasets, combining video with text or audio, offer opportunities for LLMs to align spatiotemporal and semantic reasoning, challenges remain [49, 155]. These include limited diversity in scenarios, imprecise annotations, and the difficulty of representing long-term dependencies [157, 212]. To truly evaluate and enhance LLMs' temporal understanding, future datasets must provide richer, more diverse temporal annotations and robust benchmarks across varied domains [119]. Table 3 provides a

comprehensive summary of video datasets used for tasks such as action recognition, video question answering (QA), video captioning, video retrieval, and anomaly detection.

In this work, we emphasize the role of datasets in shaping temporal understanding. We analyze the shortcomings of current video datasets, including their lack of temporal annotations, short-term bias, and limited diversity, and propose directions for improving dataset design.

**Temporal reasoning in video AI.** Temporal reasoning is critical for tasks such as action recognition, video summarization, and temporal event ordering [21, 26, 29, 44, 84, 121, 125, 156, 160–169, 212]. Classical approaches rely on recurrent neural networks (RNNs) [132], 3D convolutional networks (3D-CNNs) [18, 147], and attention mechanisms [26, 148, 164] to model temporal dependencies. More recently, transformers and temporal tokenization techniques have been explored for long-term video understanding [9, 120]. Despite these advances, the explicit modeling of abstract temporal concepts such as causality, sequence progression, and event duration remains underexplored. LLM-based approaches typically use spatiotemporal embeddings from pretrained encoders but fall short in demonstrating robust temporal reasoning capabilities, especially for complex, real-world video scenarios [157, 212].

While many studies report empirical results [7, 43, 64, 76, 106, 141, 154], we go further by offering actionable insights for advancing LLM-based temporal reasoning. These include joint training approaches, better dataset curation, and innovative multimodal fusion techniques. By addressing these gaps, our work not only advances the understanding of temporal reasoning in LLMs but also provides a roadmap for future research, distinguishing it from prior efforts[118, 142, 210] in this rapidly evolving field.

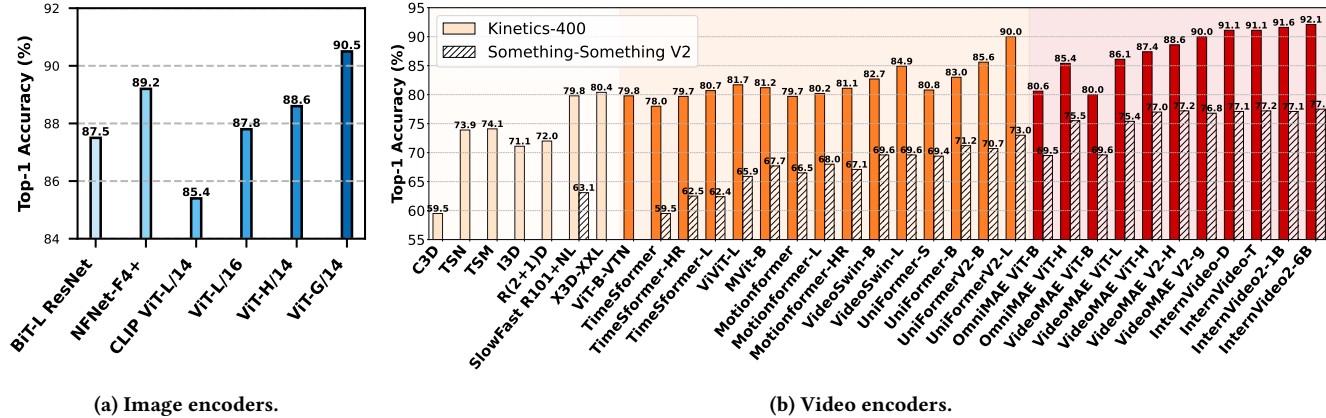

(a) Image encoders.                                                    (b) Video encoders.

**Figure 2: Performance comparison of visual encoders. (*Left*): Image classification accuracy for various image encoders pretrained and fine-tuned on the ImageNet-1K dataset. (*Right*): Action recognition accuracy for different video encoders pretrained and fine-tuned on the Kinetics-400 and Something-Something V2 datasets.**

| Task | Dataset | Year | Source | # Videos | Modality | Avg. length (s) | Temporal annotation | Description |
|---|---|---|---|---|---|---|---|---|
| | HMDB51[88] | 2011 | YouTube | 6,766 | Video | 3–4 | No | Daily human actions |
| | UCF101[137] | 2012 | YouTube | 13,320 | Video+Audio | 7.21 | No | Human actions (e.g., sports, daily activities) |
| | ActivityNet[13] | 2015 | YouTube | 27,801 | Video+Text | 300–1200 | Temporal extent provided | Human-centric activities |
| | Charades[136] | 2016 | Crowdsourced | 9,848 | Video+Text | 30.1 | Start and end timestamps provided | Household activities |
| | Kinetics-400[82] | 2017 | YouTube | 306,245 | Video | 10 | No | Human actions (e.g., sports, tasks) |
| Action Recognition | AVA[60] | 2018 | Movies | 430 | Video | Variable | Start and end timestamps provided | Action localization in movie scenes |
| | Kinetics-600[16] | 2018 | YouTube | 392,622 | Video | 10 | No | Human actions (e.g., sports, tasks) |
| | Something-Something V2[58] | 2018 | Crowdsourced | 220,847 | Video | 2–6 | Weak | Human-object interactions |
| | EPIC-KITCHENS[38] | 2018 | Participant kitchens | 432 | Video+Text+Audio | ~458 | Start and end timestamps provided | Large-scale egocentric cooking dataset |
| | COIN[143] | 2019 | YouTube | 11,827 | Video+Text | 141.6 | Start and end timestamps provided | Comprehensive instructional tasks (e.g., cooking, repair) |
| | Kinetics-700[17] | 2019 | YouTube | 650,317 | Video | 10 | No | Expanded version of Kinetics-400 and Kinetics-600 |
| | EPIC-KITCHENS-100[39] | 2020 | Participant kitchens | 700 | Video+Text+Audio | ~514 | Start and end timestamps provided | Large-scale egocentric cooking dataset |
| | Ego4D[59] | 2021 | Wearable Cameras | 3,850 hours | Video+Text+Audio | Variable | Start and end timestamps provided | First-person activities and interactions |
| | VidSitu[129] | 2021 | YouTube | 29,000 | Video+Text | ~10 | Temporal extent for events provided | Event-centric and causal activity annotations |
| | MovieQA[144] | 2016 | Multiple platforms | 408 | Video+Text | 202.7 | Start and end timestamps provided | QA over movie scenes |
| | TGIF-QA[78] | 2016 | Tumblr GIFs | 56,720 | Video+Text | 3–5 | Action timestamps provided | QA over social media GIFs |
| | MSVD-QA[187] | 2017 | YouTube | 1,970 | Video+Text | 27.5 | Start and end timestamps provided | QA for actions description |
| | MSRVTT-QA[187] | 2017 | YouTube | 10,000 | Video+Text | 15–30 | Weak | QA across diverse scenes |
| | TVQA[89] | 2019 | TV Shows | 21,793 | Video+Text | 60–90 | Start and end timestamps provided | QA over medical dramas, sitcoms, crime shows |
| Video QA | ActivityNet-QA[196] | 2019 | YouTube | 5,800 | Video+Text | 180 | Implicit (derived from ActivityNet) | QA for human-annotated videos |
| | How2QA[95] | 2020 | HowTo100M (YouTube) | 22,000 | Video+Text | 60 | Temporal extent provided | QA over instructional videos |
| | YouCook2-QA[172] | 2021 | YouCook2 (YouTube) | 2,000 | Video+Text | 316.2 | Temporal boundaries provided | Cooking-related instructional QA |
| | STAR[181] | 2021 | Human activity datasets | 22,000 | Video+Text | Variable | Action-level boundaries provided | QA over human-object interactions |
| | MVBench[94] | 2023 | Public datasets | 3,641 | Video+Text | 5–35 | Start and end timestamps provided | Multi-domain QA (e.g., sports, indoor scenes) |
| | EgoSchema[116] | 2023 | Ego4D (Wearable Cameras) | 5,063 | Video+Text | 180 | Timestamped narrations provided | Long-form egocentric activities |
| | YouCook[40] | 2013 | YouTube | 88 | Video+Text | 180–300 | Weak | Cooking instructional videos |
| | MSR-VTT[189] | 2016 | YouTube | 7,180 | Video+Text+Audio | 10–30 | Weak | General scenarios (e.g., sports, transport) |
| Video Captioning | ActivityNet Captions[86] | 2017 | YouTube | 20,000 | Video+Text | 180 | Start and end timestamps provided | Dense captions for human-centered activities |
| | VATEX[175] | 2019 | YouTube | 41,250 | Video+Text | ~10 | Weak | Multilingual descriptions with English-Chinese parallel captions |
| | HowTo100M[117] | 2019 | YouTube | 1.22M | Video+Text+Audio | 390 | Subtitle timestamps provided | Instructional video captions |
| | TVC[90] | 2020 | TV Shows | 108,965 | Video+Text | 76.2 | Start and end timestamps provided | Multimodal video captioning dataset |
| | LSMDC[128] | 2015 | Movies | 118,114 | Video+Text | 4.8 | Start and end timestamps provided | Large-scale dataset for movie description tasks |
| | DiDeMo[69] | 2017 | Flickr (YFCC100M) | 10,464 | Video+Text | 27.5 | Start and end timestamps provided | Moment localization in diverse, unedited personal videos |
| Video Retrieval | FIVR-200K[85] | 2019 | YouTube | 225,960 | Video | ~120 | Start and end timestamps provided | Large-scale incident video retrieval dataset with diverse news events |
| | TVR[90] | 2020 | TV Shows | 21,793 | Video+Text | 76.2 | Start and end timestamps provided | Video-subtitle multimodal moment retrieval dataset |
| | TextVR[185] | 2023 | YouTube | 10,500 | Video+Text | 15 | Weak | Cross-modal video retrieval with text reading comprehension |
| | EgoCVR[73] | 2024 | Ego4D | 2,295 | Video+Text | 3.9–8.1 | Weak | Egocentric dataset for fine-grained composed video retrieval |
| | Subway Entrance[2] | 2008 | Surveillance cameras | 1 | Video | 4,800 | No | Crowd monitoring for unusual event detection at subway entrances |
| | Subway Exit[2] | 2008 | Surveillance cameras | 1 | Video | 5,400 | No | Crowd monitoring for unusual event detection at subway exits |
| Anomaly Detection | CUHK Avenue[111] | 2013 | Surveillance cameras | 15 | Video | 120 | No | Urban avenue scenes with anomalies like running, loitering, etc. |
| | Street Scene[126] | 2020 | Urban street surveillance | 81 | Video | 582 | Spatial and temporal bounding boxes | Urban street anomalies, e.g., jaywalking, loitering, illegal parking, etc. |
| | XD-Violence[183] | 2020 | Movies and in-the-wild scenes | 4,754 | Video+Audio | ~180 | Start and end timestamps provided | Multimodal violence detection covering six violence types |
| | CUVA[46] | 2024 | YouTube, Bilibili | 1,000 | Video+Text | ~117 | Start and end timestamps provided | Causation-focused anomaly understanding across 42 anomaly types |
| | MSAD[212] | 2024 | Online Surveillance | 720 | Video | Variable | Frame-level annotations in test set | Multi-scenario dataset with 14 scenarios |
| Multimodal video tasks | VIDAL-10M[211] | 2023 | Multiple platforms | 10M | Video+Infrared+Depth+Audio+Text | ~20 | Weak | Multi-domain retrieval dataset |
| | Video-MME[49] | 2024 | YouTube | 900 | Video+Text+Audio | 1017.9 | Temporal ranges via certificate length | Comprehensive evaluation benchmark across many domains |

**Table 3: Comprehensive overview of video datasets across tasks.**

# 3 Analysis and Discussion

**Can LLMs understand the concept of time?** LLMs exhibit several strengths in processing temporal information. Trained on textual data containing narrations or instructions, they can infer temporal relationships through contextual cues such as "first", "then", and "after". When paired with video encoders, LLMs can process spatiotemporal embeddings, enabling tasks like action recognition and temporal event ordering.

However, LLMs lack direct temporal awareness. Standard models do not inherently model the flow of time unless explicitly trained on sequential video data. Instead, they rely on external encoders to provide temporal structure. Capturing long-term dependencies

over extended video sequences is another challenge, as LLMs often operate on tokenized inputs within limited context windows. Additionally, video encoders like 3D CNNs [18, 147] or video transformers [120, 158], which act as the "eyes" of the system, may excel in capturing motion patterns but struggle to generalize abstract temporal concepts like causality or duration.

A significant limitation lies in the representation of visual time. Unlike textual representations, visual cues require explicit modeling of motion and event transitions. This ambiguity underscores the need for improved temporal modeling in both encoders and LLMs.

**LLMs applied to videos using pretrained visual encoders.** Most existing approaches use pretrained image or video encoders

to extract visual or spatiotemporal information, rather than designing entirely new encoders (Table 1 and 2). Pretrained image encoders such as CLIP [122], ResNet [68], and ViT [148] excel in capturing spatial information, while video encoders like I3D [18], SlowFast [48], TimeSformer [9], and Video Swin Transformer [109] are widely used for spatiotemporal feature extraction. These encoders, trained on large-scale datasets such as ImageNet [41, 127] or Kinetics [17, 18, 82], are adept at learning rich feature representations, which can then be fine-tuned for specific tasks [21, 26, 44, 84, 157, 161, 165, 166, 169, 170, 212]. Notably, video encoders incorporate mechanisms to model temporal dependencies, such as optical flow [165], Taylor videos [170], and motion tracking [26, 125], addressing a challenge that LLMs alone cannot handle effectively. Figure 2 illustrates the performance of popular visual encoders on two key tasks: image classification (ImageNet-1K) and video action recognition (Kinetics-400 and Something-Something V2).

The use of pretrained visual encoders offers practical advantages. These encoders are optimized for handling visual and spatiotemporal features, reducing computational overhead and enabling faster convergence. Their modular design also allows them to function as "plug-and-play" components, seamlessly integrating with LLMs for multimodal learning. This modularity ensures that LLM-based frameworks remain adaptable and scalable across diverse applications. However, pretrained encoders are not without limitations. First, encoders trained on general datasets may underperform on domain-specific video tasks [21, 212]. Second, many pretrained encoders prioritize spatial over temporal information, necessitating additional modules, such as temporal transformers, to capture complex temporal dynamics [25, 202]. Lastly, large-scale video datasets required for training such encoders are costly to annotate, limiting their ability to encompass diverse or abstract video content.

Some recent efforts, such as DeepMind's Flamingo [5] and unified architectures like Florence [197] and OmniVL [153], explore custom encoders optimized for multimodal learning. These models aim to balance performance across multiple modalities (image, video, and text) without relying heavily on separate pretrained components [157, 161, 166].

**How encoders and LLMs interact?** Encoders play a crucial role in preprocessing video frames or sequences to extract visual or spatiotemporal features, which are then transformed into a format compatible with LLMs, often as token embeddings. For example, video transformers process sequences of video frames, while text encoders like CLIP encode textual inputs [6, 191, 201]. These features are projected into a shared representation space to align different modalities, such as visual and textual data.

Fusion mechanisms are pivotal in enabling LLMs to process multimodal inputs. Encoded features are tokenized and treated as input tokens for the LLM. Attention mechanisms, such as cross-modal attention [27, 207], are then used to integrate and relate features from different encoders. Positional embeddings encode spatial and temporal positions of video features, helping the LLM model sequence dependencies. For instance, cross-modal transformers align visual and textual representations, while LLMs refine these embeddings to generate outputs like action labels, video descriptions, or anomaly detection results [47].

Table 1 highlights the interaction/fusion mechanisms used in the latest video-LLMs under the 'Interaction / Fusion mechanism'

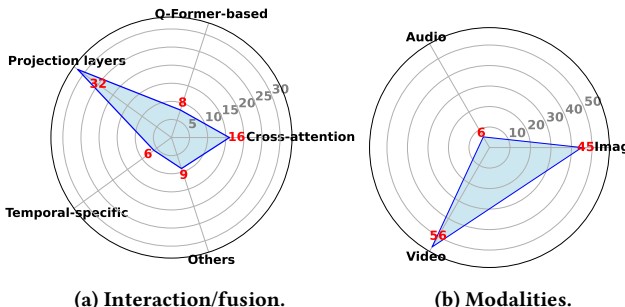

(a) Interaction/fusion.                  (b) Modalities.

**Figure 3: The distributions of interaction/fusion mechanisms and data modalities in 66 closely related video-LLMs from January 2024 to December 2024. (*Left*): Fusion mechanisms are classified into five categories: Cross-attention (*e.g.,* cross-attention modules, gated cross-attention), Projection layers (*e.g.,* linear projection, MLP projection), Q-Former-based methods (*e.g.,* Q-Former aligns multi-modal embeddings, Trainable Q-Former), Motion/Temporal-Specific mechanisms (*e.g.,* temporal samplers, scene-level sequential alignment), and Other Methods (*e.g.,* Tracklet-centric, Perceiver Resampler, MQ-Former). (*Right*): The distribution of data modalities used in these video-LLMs, with text modalities appearing across all models. Note that a model may use multiple fusion methods and/or data modalities.**

column. Figure 3a visualizes the distribution of various interaction and fusion techniques, while Figure 3b showcases the modalities used in closely related works from 2024 (notably, text modalities are consistently used across these studies). As shown in these plots, the majority of works use projection layers (*e.g.,* linear projection [94], MLP projection [50, 203, 205], semantic translators [20, 25]) and cross-attention mechanisms (*e.g.,* cross-modal attention [191], gated cross-attention [150]) to facilitate interaction between encoders and LLMs. Temporal-specific mechanisms, such as temporal sampler [202] and scene-level sequential alignment [193], are also used. Emerging video-LLMs are beginning to explore novel fusion mechanisms, including tracklet-centric approach [151], Perceiver resamplers [5, 213], and MQ-Former [152].

However, several challenges arise in this interaction. Encoders must output features in a format compatible with LLMs, requiring careful dimension alignment and embedding space mapping. Additionally, processing video data generates a substantial volume of information, which can strain the LLM's capacity. Capturing long-term dependencies across extended video sequences also remains a challenge, even with support from pretrained encoders.

**Bridging the gap between raw video data and temporal reasoning.** The interaction between LLMs and encoders helps bridge the gap between raw video data and higher-level temporal reasoning. Pretrained encoders extract spatiotemporal embeddings that encapsulate low-level motion cues, such as velocity [170] and trajectory [120], as well as higher-level temporal patterns like scene transitions and sequence progression. These embeddings provide the foundation for LLMs to interpret complex temporal concepts such as causality, event progression, and anticipation.

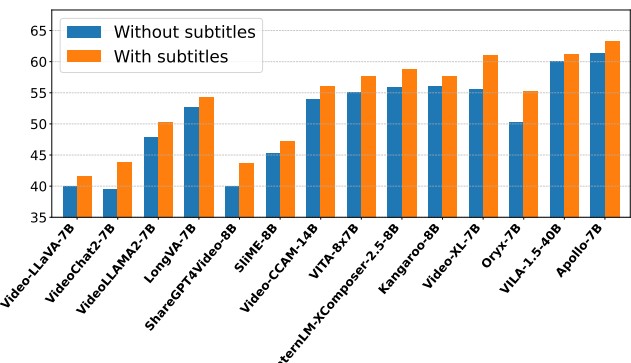

Figure 4: Performance (accuracy) comparison of recent video-LLMs on the Video-MME benchmark.

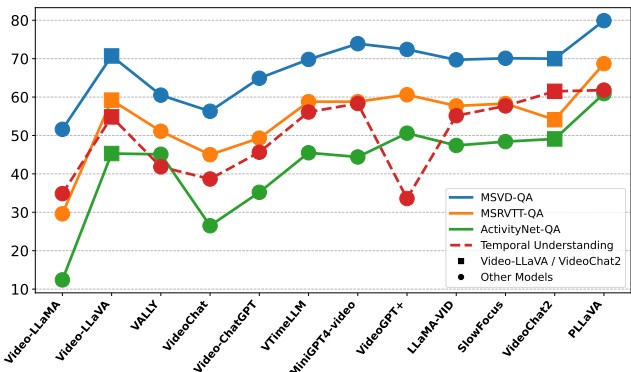

Figure 5: Performance comparison of recent video-LLMs on video QA benchmarks. Models using pretrained video encoders (*e.g.,* Video-LLaVA and VideoChat2) are marked with squares, while models using pretrained image encoders are represented by circles.

LLMs use attention mechanisms to prioritize spatiotemporal features, aligning them with semantic or task-specific contexts. Encoders often supply frame-level features as temporal tokens, enabling the LLM to model dependencies and transitions across frames. However, significant challenges remain in achieving comprehensive temporal understanding. Encoders frequently focus on short-term motion patterns, neglecting long-term dependencies [162, 167, 168]. Similarly, datasets used for encoder pretraining often lack diversity and fail to represent abstract temporal relationships effectively [157]. Table 3 provides a comparison of datasets based on average video length, data source, modalities, and the number of videos.

LLMs also face inherent limitations. Temporal embeddings require careful tokenization to preserve sequence information when input into LLMs (see Table 1). Additionally, LLMs pretrained on static datasets such as text or images may lack the dynamic reasoning capabilities required for video tasks [65]. Combining spatial, temporal, and semantic information without losing critical cues remains a complex challenge [21, 26, 125], further compounded by the computational expense of processing long video sequences while retaining both global and local temporal details.

**Video datasets: an enabler or bottleneck?** Video datasets are foundational to LLMs on video tasks, yet they often act as a bottleneck [157]. Action recognition datasets, such as Kinetics [16, 82] and Something-Something V2 [58], are effective for analyzing short-term motion patterns but lack the temporal annotations necessary for reasoning about action sequences or causal relationships. Compared to the Something-Something V2 dataset, the Kinetics datasets exhibit a stronger spatial bias, as demonstrated in [55, 57, 120]. Similarly, video QA datasets like TVQA [89] and How2QA [95] align well with LLM architectures but often rely on scripted scenarios that limit generalizability to real-world tasks.

Video captioning datasets like MSR-VTT [189] and ActivityNet Captions [86] enable multimodal learning by fusing video and text embeddings. However, captions are often superficial and fail to probe deeper temporal reasoning. Long-term video understanding datasets, such as Ego4D [59] and Charades [136], focus on extended activities and interactions, offering a richer testing ground for temporal reasoning. Nonetheless, LLMs often struggle with the scale and complexity of such datasets, given their limited context windows.

Multimodal datasets like HowTo100M [117] and COIN [143] align video content with auxiliary modalities, providing opportunities for pretraining in a multimodal setup. However, the inherent noise and lack of temporal annotations in these datasets can hinder performance. To advance temporal understanding in LLMs, datasets must evolve to include richer temporal annotations, long-term dependencies, and diverse real-world scenarios. The latest multimodal datasets include VIDAL-10M [211] and Video-MME [49].

Table 3 summarizes popular datasets used across various video tasks, including action recognition, anomaly detection, video question answering, captioning, and retrieval, as well as some recent multimodal video understanding datasets.

**State-of-the-Art video LLMs.** Recent advancements in video LLMs have significantly enhanced the processing of spatiotemporal information [22, 94, 119]. Traditional LLMs, focused on textual data, often struggle with temporal dynamics in video [106]. Models like VideoChat2 [94] and SlowFocus [119] are pushing the boundaries by integrating temporal reasoning with video analysis. VideoChat2 enables real-time multimodal dialogue, processing video sequences to answer questions about actions, events, and causal relationships, making it highly effective for interactive applications. SlowFocus improves fine-grained temporal understanding, capturing long-term dependencies and transitions within video, essential for tasks like video summarization and anomaly detection. TimeSformer [9] further refines this by using attention mechanisms to simultaneously model spatial and temporal features, enhancing video understanding in complex scenarios like action recognition.

Additionally, models, *e.g.*, VideoLLM-Online [22] and VideoBERT [139], focus on continuous learning from video streams, allowing for real-time updates and better adaptability to dynamic content. These models are crucial for applications requiring ongoing video analysis, such as surveillance, event detection, and interactive media. Flamingo [5] takes multimodal learning a step further by combining visual, textual, and temporal data, offering a more holistic approach to video processing. ActionFormer [200], on the other

hand, specializes in action recognition through long-range temporal dependencies, making it effective for tasks like sports video analysis and human-computer interaction. These advancements reflect a significant leap in video LLM capabilities, making them better equipped for real-world video understanding, interaction, and analysis. Figures 4, 5, and 6 present comparisons of recent popular video-LLMs across multiple video tasks, including multimodal video understanding, video QA, video retrieval, and video captioning. As shown in these figures, no single video-LLM excels across all tasks: (i) a comprehensive evaluation system covering all video tasks is lacking, and (ii) most video-LLMs are designed to address only a subset of these challenges.

**Fair evaluation is needed.** Evaluations and comparisons of video-LLMs are often conducted inconsistently (see Figures 4, 5, and 6), which can result in unfair assessments and misleading conclusions [118, 142]. A frequent issue is the comparison of video-LLMs, designed for multimodal reasoning, with traditional video models such as I3D [18], SlowFast [48], or Video Swin Transformer [109], which are tailored for video-specific tasks like action recognition. While traditional models excel at spatiotemporal feature extraction due to their focused design, video-LLMs must simultaneously handle visual and linguistic alignment, which adds inherent complexity to their objectives. Directly comparing video-LLMs against such specialized models is therefore not entirely fair. Instead, video-LLMs should be systematically benchmarked against other video-LLMs or multimodal frameworks to better reflect their relative strengths and limitations in tasks like video action recognition, video captioning, or video QA.

Furthermore, inconsistencies in evaluation practices exacerbate the problem. Different models are often trained and tested on varying datasets, such as Kinetics-400 [82], Ego4D [59], or HowTo100M [117], without standardized protocols for pretraining, finetuning, or testing. This creates biases in results and hampers fair comparisons. For instance, a video-LLM pretrained on massive multimodal datasets might show superior results simply due to larger data availability rather than architectural improvements. To address this, evaluations must adopt consistent and standardized benchmarks, training splits, and metrics. Frameworks that systematically assess multimodal alignment, temporal reasoning, and downstream task performance would help ensure transparency and comparability.

Finally, establishing fair evaluation practices requires prioritizing within-paradigm comparisons. For video action recognition, for example, models like VideoChat2 [94] and VideoLLM-online [22] should be compared against each other rather than with traditional video-only transformers. This approach highlights progress within the multimodal video understanding space and reveals areas for improvement, such as better temporal consistency or more efficient multimodal alignment. By addressing these challenges, fair and systematic evaluation will provide deeper insights into video-LLM capabilities and foster future advancements in the field.

**Factors driving superior performance in video LLMs.** The superior performance of certain video LLMs can be attributed to their ability to effectively integrate spatial, temporal, and semantic information, often through advanced architectures and training strategies. Models like SlowFocus [119] and VideoChat2 [94] excel by incorporating fine-grained temporal reasoning and long-range dependencies, which are crucial for understanding complex video

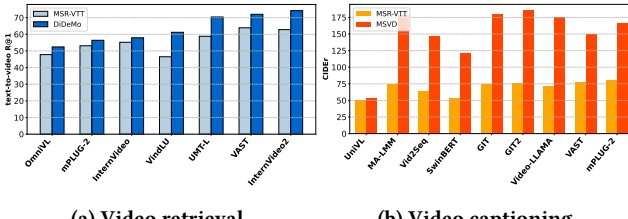

(a) Video retrieval.                    (b) Video captioning.

**Figure 6: Performance comparison of recent video-LLMs on (a) video retrieval and (b) video captioning benchmarks.**

dynamics such as event progression and causal relationships. The use of hierarchical or multi-level attention mechanisms, seen in models like TimeSformer [9] and VideoLLM-Online [22], enables them to capture both short-term motions and long-term narrative structures, addressing the temporal gaps that limit earlier models.

Additionally, models that use large-scale, multimodal pretraining on diverse video datasets, such as Flamingo, benefit from the cross-domain knowledge transfer between visual, textual, and temporal modalities, leading to a more holistic understanding of video content. These innovations enable the models to generalize better across different video tasks, including action recognition, video captioning, and dynamic scene interpretation.

## 4 Future Directions

Building on the preceding analysis and discussion, we outline below several promising future research directions for those interested in advancing video LLMs.

**Overcoming dataset challenges for LLMs.** Datasets remain a critical bottleneck in advancing LLM-based video systems. Addressing their limitations requires both creative solutions and resource investments:

i. Temporal annotations and structure: Enriching datasets with temporal annotations, such as event order, duration, and causal relationships, is essential. Techniques like crowd-sourced annotations, AI-assisted labeling, or synthetic data generation could address the scarcity of such datasets.

ii. Balancing scale and quality: While large-scale datasets like HowTo100M offer broad coverage, they often include noise and irrelevant data. Future efforts should focus on curating high-quality, balanced datasets that prioritize diversity and accuracy without sacrificing scale. Semi-supervised and unsupervised learning approaches could also mitigate the need for large annotated datasets.

iii. Addressing short-term bias: The dominance of short video clips in existing datasets limits models' ability to reason over extended sequences. Introducing datasets with long-term dependencies, such as episodic or procedural videos, would help train models to understand overarching narratives and transitions.

iv. Expanding dataset diversity: Current datasets often focus on narrow domains, such as sports or cooking, which restricts model generalizability [157]. Diverse datasets encompassing a wider range of cultural contexts, real-world scenarios, and tasks are essential to improve performance across applications. Collaborative initiatives between academic and industry stakeholders could accelerate the development of such datasets.

v. Improving multimodal alignment: Misalignment between video frames and associated text or audio annotations introduces inconsistencies that can degrade learning quality. Future work could explore more precise alignment methods [162, 163, 167, 168], such as neural alignment models or reinforcement learning frameworks, to ensure that temporal and semantic signals are accurately correlated during training.

**Enhancing temporal understanding.** To improve temporal reasoning, joint training of encoders and LLMs is a promising direction. Models co-trained on datasets with temporal reasoning tasks can develop a deeper understanding of complex time-related concepts such as causality, event sequencing, and duration. Architectures like temporal transformers, recurrent neural networks, or hybrid systems that combine hierarchical and sequential processing should be further explored to handle both short-term dynamics and long-term dependencies in video data.

Explicit supervision for abstract temporal concepts through enriched annotations is another critical step [44]. Annotated datasets with detailed temporal labels, covering relationships, transitions, and event causality, can significantly boost the temporal reasoning capacity of these systems. Furthermore, multimodal training on datasets that integrate video, text, audio, and temporal metadata could align spatiotemporal and semantic representations more effectively, enhancing the model's ability to reason about time in real-world scenarios [44, 125, 212].

**Multimodal LLMs for holistic temporal reasoning.** The development of truly multimodal LLMs capable of holistic temporal reasoning requires a synergistic interplay between spatiotemporal and semantic understanding. Future research should explore adaptive attention mechanisms that dynamically weigh temporal, spatial, and textual information based on context. Advances in memory-efficient architectures and progressive learning strategies could enable LLMs to process longer sequences without losing critical details.

Additionally, fine-tuning LLMs on domain-specific datasets or tasks, such as medical video analysis or video-based social interaction studies [36, 79], could expand their applicability and temporal reasoning depth. Transfer learning approaches [206], where models pretrained on general datasets are fine-tuned for specific temporal reasoning tasks, can also be effective in reducing resource demands.

Other modalities, such as depth videos [156, 159] and motion-specific data like skeletons (useful for analyzing human-related movements) [162–164, 167, 168], optical flow [165], and Taylor videos [170], can significantly enhance the performance of LLM frameworks in video processing tasks when their pretrained models are incorporated. Depth videos capture three-dimensional spatial information, offering a richer understanding of scene geometry. Skeleton data focuses on joint movements, making it particularly effective for applications like action recognition and gesture analysis. Optical flow and Taylor videos excel in capturing frame-to-frame motion changes, providing detailed temporal cues essential for understanding dynamic content.

By integrating these diverse modalities, LLMs can achieve a more comprehensive representation of motion dynamics and spatial structures, broadening their applicability to complex video-based challenges. Moreover, the inclusion of learned video motion

prompts [26], which highlight relevant movements within a scene, introduces a novel modality that further refines the system's ability to process and interpret intricate video content.

**Advancing visual encoders for multimodal learning.** While most current LLM-based video systems use pretrained encoders for their efficiency and robust feature extraction, there is significant potential in designing novel encoders optimized specifically for multimodal learning. These encoders should aim to seamlessly integrate spatial, temporal, and semantic information into a unified framework, reducing the reliance on modular preprocessing steps. Future research could explore adaptive encoder architectures that dynamically adjust to varying video characteristics, such as scene complexity or temporal dynamics. Additionally, encoders tailored for specific domains, like healthcare, education, or autonomous driving, could enhance the accuracy and relevance of multimodal systems in specialized applications.

**Ethical and practical considerations.** As LLM-based video systems advance, addressing ethical concerns becomes increasingly important [170, 212]. Ensuring fairness and avoiding biases in datasets, particularly regarding cultural and contextual diversity, will be critical. Practical considerations like energy efficiency and the environmental impact of large-scale training should also guide future research. Lightweight models or distillation techniques could balance performance with computational sustainability. Addressing these directions, researchers can unlock the full potential of LLMs in video-based applications, driving advancements in temporal reasoning, multimodal understanding, and real-world usability.

## 5 Conclusion

This work critically examines the temporal reasoning capabilities of large language models (LLMs) in video processing, identifying significant limitations in both models and datasets. While LLMs paired with pretrained visual encoders have achieved success in tasks such as action recognition, anomaly detection, and video summarization, they fall short in understanding long-term temporal dependencies. This stems from the encoders' focus on short-term patterns, fragmented temporal cues, and challenges in aligning spatial, temporal, and semantic information. Additionally, existing datasets lack explicit temporal annotations, often focus on short clips over long sequences, and struggle with diversity and multimodal alignment, further hindering progress.

To unlock the full potential of LLMs in video processing, future research must address these gaps. This includes designing integrated frameworks to jointly train encoders and LLMs on temporal reasoning, enriching datasets with detailed annotations for long-term dependencies, and creating innovative architectures that fuse spatiotemporal and semantic information. By addressing these challenges, we can pave the way for systems that not only excel in video analysis but also advance broader applications requiring robust temporal comprehension.

## Acknowledgments

Xi Ding, a Research Assistant with the Temporal Intelligence and Motion Extraction (TIME) Lab at ANU, contributed to this work. This research was conducted under the supervision of Lei Wang.

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
