# OpenReview forum: "Do Language Models Understand Time?"
_ACM.org/TheWebConf/2025/Workshop/TIME — TIME 2025 Oral_

### Official Review · Reviewer_MVqB · 2025-01-15
**Review for #2**

**Rating:** 6
**Confidence:** 4

**Review:**

The paper provides a comprehensive recent review of LLM applications in video processing, analyzes the limitations of existing approaches, and suggests potential future directions. The content is substantial, but the following aspects could be improved:

-	The title might be somewhat general and vague. As noted in the paper, LLMs struggle with understanding long-term temporal dependencies. Is their performance in short-term scenarios satisfactory? It would be helpful to clarify this point in the content and title. Otherwise, it can confuse people.

-	Figure 1 illustrates an application in video captioning, but it appears to have little connection to the subsequent content of the paper. Since the paper functions more as a review and does not aim to address this specific problem presented in the example, it might be more appropriate to replace the figure with one that highlights a broader range of applications or challenges.

-	Including a figure and paragraph to outline the content and the relationships between different sections could make the structure clearer and help readers grasp the main points more effectively, given the extensive scope of the content.

---

### Official Review · Reviewer_NjhZ · 2025-01-20
**Do Language Models Understand Time?**

**Rating:** 8
**Confidence:** 5

**Review:**

**Summary**

This paper provides a comprehensive exploration and analysis of whether LLMs understand the concept of time and how they reason about temporal relationships in videos. The authors identify that the limitations hindering time understanding lie in the interaction between LLMs and the pretrained visual encoder, as well as dataset bias due to a lack of temporal annotations. Based on these analyses, the paper proposes several promising directions for improvement to advance the temporal understanding of LLMs.

**Pros**:

1.	Good Writing. This paper is well-written with a clear and concise presentation, making it easy for readers to follow and understand the main arguments and findings.
2.	Significance. The paper offers an in-depth analysis and discussion on the temporal understanding capabilities of LLMs. This topic is highly significant, as it addresses critical challenges in the design and development of contemporary LLMs, particularly for applications requiring robust reasoning about time.
3.	Comprehensiveness. This paper provides a thorough and comprehensive review of existing LLMs, including some of the most recent advancements, in understanding time.

**Cons**:

Although this paper offers several promising future directions, the authors do not provide validated methods for these specific directions, which may impact the overall quality and practical contribution of the paper.

---

### Official Review · Reviewer_XbTM · 2025-01-23
**Comprehensive review lacking novel solution**

**Rating:** 7
**Confidence:** 2

**Review:**

his paper investigates whether large language models (LLMs) can comprehend temporal concepts, specifically sequential information in videos. In my opinion, this topic should strictly fall within the domain of vision-language models (VLMs). The paper provides a comprehensive summary of related work, along with thorough analysis and discussion of future directions. However, it does not present a novel solution to the problem. Despite this limitation, I believe it is worth reading, which is why I have given it a relatively positive score. I am uncertain whether it fully meets the specific criteria of this workshop.

---

### Official Review · Reviewer_oWqx · 2025-01-24

**Rating:** 6
**Confidence:** 3

**Review:**

This paper presents an interesting analysis of the limitations of current Large Language Models (LLMs) in understanding temporal concepts within videos. The authors rightly point out the over-reliance on short-term dependencies in existing video-LLM interactions and the lack of robust temporal modeling.  However, the paper seems to fall short in providing concrete evidence to support its claims.

While the proposed solutions of co-evolving LLMs and encoders, enriching datasets with explicit temporal labels, and developing new architectures for spatio-temporal-semantic reasoning are promising avenues, they lack the necessary detail and rigor.  The authors need to go beyond theoretical analysis and offer Quantitative Evaluation, Implementation Details and Empirical Validation

In its current form, the paper reads more like a position paper than a robust research contribution.  By incorporating concrete implementations,  well-defined metrics, and empirical validation, the authors can significantly strengthen their arguments and contribute meaningfully to the field.

---

### Meta-Review · Area_Chair_79rW · 2025-01-25

**Recommendation:** Accept (Oral)
**Confidence:** 4

**Metareview:**

The paper explores underexplored and complex topics i.e. how LLMs handle time in video tasks. It does a great job of identifying gaps in current approaches and suggesting ways to fix them. While it’s mostly theoretical, the level of analysis is solid, and the ideas feel well thought out.

Clarity and Originality:
The writing is straightforward and easy to follow. The structure flows well, and most concepts are explained clearly. That said, a few technical parts, like spatiotemporal embeddings, could be made simpler or maybe even paired with visuals to help readers get the idea faster. The focus on temporal reasoning is underexplored, and it’s nice to see an underexplored area being addressed. While the suggestions, like improving datasets or joint training, aren’t groundbreaking, they’re practical and useful steps forward.

Pros: The paper highlights real gaps in how LLMs process time, especially with datasets and long-term dependencies. It also gives clear, actionable ideas for moving the field forward, which makes it valuable for researchers.
Cons: The lack of experimental results makes some of the claims feel a bit untested. It also doesn’t spend enough time on practical challenges, like how to handle scalability or computational costs.

Overall: This paper has strong ideas and clear potential to make an impact. It’s engaging and forward-thinking.

---

### Decision · Program_Chairs · 2025-01-26

**Decision:**

Accept (Oral)

**Comment:**

This paper was reviewed by four reviewers, all of whom agreed that it is an interesting and comprehensive review, offering in-depth analysis.

The area chair also acknowledged the paper's strong ideas and clear potential to make a significant impact.

They found the work to be engaging and forward-thinking.

The program chair concurs with the area chair's recommendation and endorses this paper for an oral presentation.

For the camera-ready version, please revise your paper according to the feedback provided by the reviewers.

Workshop papers must be written in English, follow a double-column format, and comply with the [ACM template](https://www2025.thewebconf.org/short-papers) and formatting guidelines. The template is also available in [Overleaf](https://www.overleaf.com/latex/templates/association-for-computing-machinery-acm-sig-proceedings-template/bmvfhcdnxfty). For authors using Microsoft Word, the Word Interim Template is recommended.

Camera-ready versions of accepted papers can and should include all information to identify authors, and should acknowledge any funding received that directly supported the presented research.

In addition, ensure that the DOI (to be provided by the PCs at a later stage) is included, and cite the workshop (to appear) using the following reference:

```
@inproceedings{time2025,
  title={TIME 2025: 1st International Workshop on Transformative Insights in Multi-faceted Evaluation},
  author={Lei Wang and Md Zakir Hossain and Syed Islam and Tom Gedeon and Sharifa Alghowinem and Isabella Yu and Serena Bono and Xuanying Zhu and Gennie Nguyen and Nur Haldar and Seyed Jalali and Abdur Razzaque and Imran Razzak and Rafiqul Islam and Shahadat Uddin and Naeem Janjua and Aneesh Krishna and Manzur Ashraf},
  booktitle={ACM Web Conference Workshop},
  year={2025}
}
```

Please note that at least one in-person registration is required for each accepted workshop paper to be included in the Companion Proceedings of WWW 2025. All accepted papers must be presented at the conference. Papers not presented (no-shows) may be withdrawn from the companion proceedings. Presentations will be conducted in two formats: oral and poster.

The camera-ready deadline for workshop papers is 7 February 2025 (AoE).